# Development of a Real-Time Surface Solar Radiation Measurement System Based on the Internet of Things (IoT)

**DOI:** 10.3390/s21113836

**Published:** 2021-06-01

**Authors:** Álvaro B. da Rocha, Eisenhawer de M. Fernandes, Carlos A. C. dos Santos, Júlio M. T. Diniz, Wanderley F. A. Junior

**Affiliations:** 1Academic Unit of Mechanical Engineering, Federal University of Campina Grande, Campina Grande, Paraíba 58429-900, Brazil; alvarobarbosa2@hotmail.com (Á.B.d.R.); eisenhawer@ee.ufcg.edu.br (E.d.M.F.); wanderley.ferreira@ufcg.edu.br (W.F.A.J.); 2Academic Unit of Atmospheric Sciences, Federal University of Campina Grande, Campina Grande, Paraiba 58429-900, Brazil; 3School of Apprentices-Sailors of Pernambuco, Navy of Brazil, Olinda, Pernambuco 53110-901, Brazil; julio.tavares@marinha.mil.br

**Keywords:** surface solar radiation, low-cost sensor, real-time, pyranometer, Internet of Things

## Abstract

The determination of the levels of solar radiation incident on the terrestrial surface (W·m^−2^) is essential for several areas such as architecture, agriculture, health, power generation, telecommunications, and climate forecasting models. The high cost of acquiring and maintaining radiometric equipment makes it difficult to create and expand monitoring networks. It contributes to the limited Brazilian radiometric network and affects the understanding and availability of this variable. This paper presents the development of a new surface solar radiation measurement system based on silicon photodiodes (Si) with a spectral range between 300 nm and 1400 nm incorporating Internet of Things (IoT) technology with an estimated cost of USD 200. The proposed system can provide instantaneous surface solar radiation levels, connectivity to wireless networks and an exclusive web system for monitoring data. For the sake of comparison, the results were compared with those provided by a government meteorology station (INMet). The prototype validation resulted in determination coefficients (R^2^) greater than 0.95 while the statistical analysis referred to the results and uncertainties for the range of ±500 kJ·m^−2^, less than 4.0% for the developed prototypes. The proposed system operates similarly to pyranometers based on thermopiles providing reliable readings, a low acquisition and maintenance cost, autonomous operation, and applicability in the most varied climatological and energy research types. The developed system is pending a patent at the National Institute of Industrial Property under registration BR1020200199846.

## 1. Introduction

Solar radiation per unit area (W·m^−2^), or irradiance, is an essential variable for several applications including renewable energies, agriculture, health, engineering, and architecture. Two main methods have been used to obtain surface solar radiation, surface monitoring by instruments and that based on satellite images [1,2,3,4]. Both methods can have a high ability to provide radiometric data but with several technical limitations. Solar irradiance based on a planar surface of monitoring the surface instrument has the disadvantage of being expensive to implement because pyranometers are used. On the other hand, the solar irradiance method based on satellite images presents technical limitations as it is incapable of providing real-time data [1].

The pyranometer is a high-precision instrument dedicated to measuring solar radiation flux density on a horizontal surface based on a thermopile sensor or silicon semiconductor (Si). The two types of pyranometers differ in spectral sensitivity, time response and implementation cost [2,3,4]. Pyranometers based on thermopiles have a spectral sensitivity of between 300 nm and 2800 nm with a time response higher than 10 s and a cost that can surpass USD 10,000. Pyranometers based on silicon semiconductors have a selective spectral sensitivity from 350 nm and 1100 nm with a time response of 10 μs and lower cost [5]. However, semiconductor-based pyranometers require a significant investment to implement and expand radiometric monitoring networks. In Brazil, the cost of implementing an essential infrastructure for monitoring solar irradiation has become quite limited for the national radiometric network, affecting the understanding of the behavior and availability of these crucial environmental data. The implementation cost of solar irradiation measurement systems has imposed restrictions as it is a point value [1,5,6,7,8,9,10].

The development of systems for the continuous measurement of solar irradiation has been increasingly recurrent. Nwankwo et al. [11] developed a pyranometer based on Si semiconductors. The proposed system had minimal hardware and was based on a digital multimeter. The irradiance was determined using calibration factors with no data transmission. Rus-Casas et al. [1] presented an electronic device capable of measuring solar radiation using Internet of Things (IoT) concepts to transmit the data. The developed system was aimed at photovoltaic applications, measuring the diffuse and global components of solar radiation. This system utilized low-cost microcontrollers to obtain global solar radiation values from sensors and transmit them to cloud-based data.

Martínez et al. [5] developed a pyranometer based on Si semiconductors with a spectral sensitivity in the range of 400 to 750 nm. The system featured hardware based on a programmable interface controller (PIC) featuring a heating system, communication via the serial port for the transmission of the collected data and level adjustment. In order to correct the effect of the movement of the solar disk, a diffuser made of polytetrafluoroethylene (PTFE) was implemented. Balan et al. [12] developed a system to monitor solar radiation based on a microcontroller and data transmission via a wireless network. The transmitted data was stored in a structured database, allowing immediate consultation.

Thus, commercial solutions that combine measurement irradiation with wireless data transfer technology are scarce and expensive, leading to the development of systems that combine the continuous monitoring of surface solar irradiation with the IoT, thus reducing cost and uncertainty [13,14]. The IoT corresponds with the set of objects, generically treated as “things”, with software, sensors, and protocols to connect to the world wide web, allowing a remote interaction with the physical environment [15,16,17]. The IoT enables intelligent devices to communicate with high latency wireless networks, offering reliability, security, and privacy of the transmitted data [18,19,20].

This study presents the development, implementation, and validation of the preliminary results for a system that allows a remote measurement of surface solar irradiation in real-time. The proposed system uses IoT technology to safely enable low-cost, intelligent, and reliable measures and provide radiometric data from multiple web platforms. The study’s contribution is to present a temperature-insensitive smart electronic device that can send data to web interfaces and reduce the pyranometers based on the cost of semiconductor junctions.

## 2. System Architecture

### 2.1. Conceptual Design

The proposed system for the real-time measurement of surface solar irradiation is presented in Figure 1. It comprises an irradiation sensor, an indicator for status/alert messages, a signal conditioning circuit, and a low-consuming energy microcontroller. The microcontroller Tensilica Xtensa LX106 implements the MQTT (message queuing telemetry transport) protocol for data transmission.

The prototype has a rectangular external body 125 mm long × 85 mm wide × 70 mm high) and 3 mm thick polyethylene. The case has layers of acrylic paint to increase the resistance to ultraviolet (UV) radiation. It has hygroscopic salt inside to prevent condensate formation and has an adjustable base allowing the system to be installed on pedestals or horizontal surfaces.

Figure 2 represents the hardware structure of the proposed system. The principle of the operation and resources inherent to the prototype system will be addressed in Section 2.2, Section 2.3, and Section 2.4.

### 2.2. Hardware

The electronic hardware is composed of a microcontroller, a signal conditioning circuit, and an irradiation sensor. The microcontroller runs the developed firmware to control all processes necessary to measure surface solar irradiation with an autonomous operation and self-supervision.

#### 2.2.1. Radiation Sensor

The radiation sensor (Figure 3) consists of a diffuser (A), a photodiode (B), a temperature sensor (C) and a housing (D) that contains all of the elements mentioned.

The diffuser (A) is located at the top of the prototype, acting as a directional optical window made of polytetrafluoroethylene (PTFE). The selection of the material was based on the optical properties of constant transmittance for the entire range of spectral sensitivity of the photodiode and resistance to UV radiation, providing a response to the displacement of light with losses of less than 5% and a dimensional stability for the chosen geometry [5,21,22,23]. The diffuser has beveled edges at 45° and a diameter that allows the photodiode a semi-angle of sensitivity (α) greater than 65°, offsetting the displacement of the light rays at low altitudes [5,22].

The radiation sensor is made up of a Vishay BPW34 model photodiode. It was chosen based on the characteristics listed in Table 1 from a technical analysis of the most popular commercial models.

The photodiode output signal (ΔI_ph_) presents the relationship between the incident surface solar irradiance (R), the active surface area (A) and the photodiode’s responsiveness (β). The estimation of the output signal is obtained by Equation (1), based on the irradiance of 1000 W·m^−2^ and the photodiode characteristics listed in Table 1.
(1)$${\Delta I}_{\mathrm{ph}}=R\cdot A\cdot \beta .$$

#### 2.2.2. Signal Conditioning Circuit

The signal conditioning circuit (Figure 4) converts the photodiode output signal (ΔI_ph_) into a proportional voltage signal (V_out_). An instrumentation amplifier circuit implemented the conditioning circuit (U1–U3) and buffer (U4). The buffer (U4) provides electric isolation from the photodiode and keeps the output signal V_out_ stable [25,26].

The topology of the instrumentation amplifier was modified, turning it into a combination of two negative feedback trans-impedance amplifiers (U1 and U2) connected to a differential amplifier (U3). The topology obtained allows the polarization current of the photodiode (ΔI_ph_) to flow simultaneously in the two feedback resistors (RF). Thus, the output voltage (V_out_) is proportional to the product between the photodiode current (ΔI_ph_) and the resistors RF, R6 and R4. Equation (2) presents the relationship between the output voltage (V_out_) and photodiode current (ΔI_ph_).
(2)Vout=∆Iph·2·RF·(R6R4).

Figure 5 shows the relationship between V_out_ and ΔI_ph_. The values of RF, R4 and R6 resistances are 22 Ω, 4.7 kΩ and 6.8 kΩ, respectively. The resistor’s values were determined based on the estimation of ΔI_ph_ necessary to obtain a maximum output voltage of 400 mV (V_out_).

#### 2.2.3. Microcontroller

A low-energy microcontroller ESP8266 with a Tensilica Xtensa LX106 32-bit processor with a maximum clock of 160 MHz is used to implement the proposed solution. It presents the following resources: 64 KB RAM, 512 KB flash memory, GPIOs connectors, an internal 10-bit resolution analog-to-digital converter (ADC), 3.22 mV sensitivity and a maximum sampling rate of 80 kHz. The microcontroller’s inputs and outputs are designed for operating at voltages from 0 V to 3.3 V. The system is connected to an external 18 W DC power supply.

The microcontroller (Figure 6) has an integrated radio transceiver integrated with the IEEE 802.11b/g/n standard for a connection speed up to 2.4 GHz. The radio transceiver operates with a built-in ShockBurst™ protocol, allowing the microcontroller to maintain wireless connections to TCP/IP networks. In addition, it supports WPA/WPA2 safe communication and sets up a connection with authentication or communication encrypted by a token [27,28].

#### 2.2.4. Data Storage

The hardware presents a datalogger that stores the collected data in a removable flash memory. The datalogger has a storage capacity of 4 GB, allowing for consecutive data storage up to four years or 10^6^ cycles. The datalogger prevents lost data when the communication between the microcontroller and the wireless network is interrupted. The output files use extensions .txt or .csv with an ASCII writing pattern stored at 60 s intervals. The stored values correspond with the voltage level (mV), radiation level (W·m^−2^), sensor temperature (°C) and alerts generated by the microcontroller.

### 2.3. Firmware

In order to perform a system operation to regulate surface radiation and data storage, the firmware was developed in C++ language and is shown in Figure 7. The firmware controls and executes all of the processes essential for a hardware operation (threads). The threads can generate signal statuses or alerts by a visual indication of an RGB LED. The signal statuses or alerts are listed in Table 2.

Thread 01 is responsible for the device′s startup and communication with local internet networks. It identifies available wireless networks within 50 m of the device, connects the device and synchronizes it with the server. Thread 02 verifies the operation and availability of data storage space in the datalogger. Thread 03 is a self-supervising mechanism, checking the connection between the device and the wireless network with a data storage capacity while Threads 04 and 05 occur. Thread 04 is responsible for the process of reading the signal from the conditioning circuit, obtaining the irradiation value, and requesting data transmission. Thread 05 transmits data directly to the server even on limited networks.

#### 2.3.1. Data Transmission Protocol

The protocol implemented for data transmission to the server is the MQTT, a flexible network model used to send information between devices and servers. This protocol is compatible with most network protocols, allowing for orderly connections without data loss on low-power devices and wireless networks with a limited speed and high latency [29,30,31,32,33,34,35,36].

#### 2.3.2. Adjust of Radiation Sensor Output

The output signal of the conditioning circuit (V_out_) can be affected by the photodiode temperature (T) and by the average trajectory of the solar disk. In order to mitigate those effects, it has implemented correction functions based on the methodologies of King et al. [36], Kern [37] and Myers [38] and Mcarthur et al. [39].

A temperature sensor (DS18B20) was used to correct the influence of temperature (°C) on the photodiode current output, keeping readings stable during the operation. A linear and dimensionless correction function *f*(*T*) (Equation (3)) is applied. The coefficients *θ* and *τ* are 8 × 10^−3^ °C^−1^ and 0.998, respectively.
(3)f(T)=θ· T(°C)+τ.

The function for the trajectory correction of the solar disk was based on Fanner et al. [40] and Brooks [41] (Equation (4)). It is modeled as a cosine function that uses the local time when the device is installed and is valid for the active photoperiod with time (h) in the range from 330 min to 1050 min. The coefficients used in function *g*(*h*) are γ = 2.6762 and ϖ = 3.9 × 10^−3^ min^−1^, respectively. The corrected voltage of the conditioning circuit (V_out_corr_) is expressed by Equation (5).
(4)$$g\left(h\right)={\left(\cos\left(\gamma -\varpi h\left[\min\right]\right)\right)}^{-1}$$
(5)$$V_{\mathrm{out}\_\mathrm{corr}}\left[\mathrm{mV}\right]=\frac{V\left[\mathrm{mV}\right]}{g\left(h\right)\cdot f\left(T\right)}\;\to \;V_{\mathrm{out}\_\mathrm{corr}}\left[\mathrm{mV}\right]=\frac{V\left[\mathrm{mV}\right]}{\theta T\left[{}^{\circ}C\right]+\tau }\cdot \cos\left(\gamma -\varpi h\left[\min\right]\right).$$

### 2.4. Data Visualization System

Figure 8 shows the user interface developed for data visualization. It shows the geographic location of the measurement, sensor number, time data, instantaneous surface irradiation value, the time elapsed since the last update and the download option. The user interface is available at https://www.radiacaoagora.tk/, (accessed on 27 April 2021) which uses an API of the platform ThingSpeak. A cloud computing system performs the procedures for collecting, processing, storing, exhibiting, and automatically exporting the measured data to the website and users.

### 2.5. Prototypes

The experiments were carried out with two prototypes (P01 and P02), which were calibrated and tested in Campina Grande/Paraíba in the northeast region of Brazil (NEB) at 512 m above sea level with average annual temperatures of 23.5 °C and annual rainfall between 300 and 1200 mm [42,43,44]. The prototypes were installed in an urban area, 3 km away from the reference weather station (Figure 9). The installation of the devices maintained a maximum inclination of 2.0 °C in relation to the horizontal plane and was free of shading on the devices.

### 2.6. Calibration

The calibration of the prototypes was carried out from 21–25 October 2019. The calibration procedure for the prototypes followed the ISO 9847 standard using a standard Lambrecht secondary pyrometer, model 16106, for a data acquisition rate of 1 min. The calibration occurred at the Brazilian National Institute of the Semi-Arid (INSA) facilities, Campina Grande, Brazil.

The data processing in the calibration data employed a K-means clustering algorithm [45], which analyzed and modeled the collected data, minimizing dispersion effects because of a variation in the atmospheric transmittance and synchronization of the data acquisition time between the prototype and the reference pyranometer.

### 2.7. Data and Experimental Area

A weather station′s official data of surface solar irradiation were obtained at the Brazilian Agricultural Research Company (Embrapa-Algodão). The weather station has a class 2 standard pyranometer connected to the Brazilian National Institute of Meteorology (INMet) and a unit of surface solar irradiation given in kJ·m^−2^.

### 2.8. Data Analysis

A data analysis compared the values provided by the prototypes and the reference data from Embrapa/INMet, which were analyzed using the statistical techniques of Reda [46], Miot [47], Diniz [48] and Tiepolo [49]. Data processing followed the statistical parameters of coefficients of determination (R^2^), Pearson’s correlation coefficient (r), mean absolute percentage error (MAPE), root mean square of relative errors (rRMS), index of agreement (d) of Willmott et al. [50] and confidence index (c). In order to classify the parameters, Table 3, Table 4 and Table 5 were used as references. In addition, the relative frequencies of radiometric divergences were analyzed and presented as histograms and a probability density function (PDF).

## 3. Results and Discussion

### 3.1. Sensor Output Signal Behavior

Figure 10 shows the voltage measurements (mV) provided by prototypes P01 and P02 in the conditions of low (insolation less than 4 h), medium (insolation between 4 and 8 h) and high atmospheric transmittance (insolation greater than 8 h).

The differences identified between the signals provided by the prototypes were due to the non-homogeneity of atmospheric phenomena within the same region, inducing the variation of readings [52]. It was observed that the behavior of the sensor voltage followed the surface solar irradiation curve at all times of the day, demonstrating that the proposed system was unaffected by variations in ambient temperature. The thermal insensitivity was caused by the real-time correction of the sensor voltage as a function of its instantaneous temperature.

### 3.2. Radiation Sensor Calibration

The data obtained during the calibration period were registered on a Cartesian plane. The *x*-axis corresponded with the system response (corrected voltage) and the *y*-axis was the value of the reference solar irradiation. At the end of the calibration period, 3200 ordered pairs were collected for each prototype. The calibration process was based on the high transmittance model. The results were treated with a k-means clustering algorithm to reduce the impact of outliers and obtain curves more representative of the data’s average behavior.

Figure 11 depicts the relationship between the reference pyranometer’s radiometric reading and the system response after data processing with the k-means algorithm.

The linear regression and statistical indices were calculated to validate the process to verify the degree of agreement between the experimental and reference data. Table 6 presents a summary of the statistical analysis for prototypes P01 and P02.

The classification for the statistical indices for the proposed prototypes occurred according to Table 3 and Table 4, indicating that the recurrence equation provided experimental safety evidence with concise solar irradiation data according to the voltage levels. The behavior of the calibration curves was similar to that presented by Lave et al. [53] and Rus-Casas [1], which obtained a correlation coefficient (R) of approximately 0.98.

### 3.3. Solar Irradiance Analysis

The proposed system results are presented as a time series in Figure 12. The system was tested for 200 days (4813 h) during the year 2020. However, it shows only the first 40 days (963 h) as the pattern was repeated for the following days.

The comparison between the reference surface solar radiation data (INMet) and the experimental results for prototype P01 is presented in Figure 13. The R^2^ value for prototype P01 was 0.962. Figure 14 compares the reference surface solar radiation data (INMet) and the experimental results of prototype P02, showing an R^2^ value of 0.958. The values obtained for R^2^ were independent of the prototype such as those obtained by Balan et al. [12] (R^2^ = 0.950) and by Shafa et al. [3] (R^2^ = 0.94).

On the other hand, the mean percentage absolute error (MAPE) obtained for prototypes P01 and P02 were −3.82% and −2.19%, respectively. The MAPE assessment considers the accuracy based on the residuals, indicating the accuracy of the forecast of the linear model. The MAPE value for the prototypes was close to the values obtained by Kim et al. [8], Dunn et al. [54] and Tohsing et al. [55] of 3.99%, 4.60% and 3.00%, respectively.

When analyzing the rRMS, it was observed that prototypes P01 and P02 presented values of 0.287 and 0.294, respectively. According to the rRMS classification (Table 6), this metric indicated an acceptable estimation of the potential for daily global radiation. High values for r, d and c were also observed for prototypes P01 and P02. For prototype P01, the values were 0.981, 0.960 and 0.941, respectively. For prototype P02, these parameters were 0.979, 0.957 and 0.937, respectively.

Figure 15 shows the scatter graph between the measured values of prototypes P01 and P02.

Based on Figure 15, it could be noted that the results were concise to each other, presenting a coefficient of determination (R^2^) equal to 0.987, a MAPE of 3.25 × 10^−6^, an rRMS of 3.25 × 10^−5^ and a c of 0.980. The values obtained for comparing the prototypes’ responses showed that they provided the measurement of surface solar radiation in a very similar and precise manner although they presented different correlation equations.

The PDF function was used to model the differences between the official data and the experimental measurements (Figure 16). It identified the probability that the prototype’s response presented divergences within the range (−500 kJ·m^−2^; 500 kJ·m^−2^). Additionally, it estimated the probability of the prototypes presenting errors only within the specified range, keeping the uncertainties of measurements under those expected.

According to Figure 16a, the frequency distribution characteristics for prototype P01 had a mean (µ) of 37.16 kJ·m^−2^ and a standard deviation (δ) of 276.92 kJ·m^−2^. The results provided by system P01 had a confidence level of 90.53% of the divergences located in the range of (−500 kJ·m^−2^; 500 kJ·m^−2^). On the other hand, the frequency distribution for prototype P02 (Figure 16b) presented a mean (µ) value equal to 21.29 kJ·m^−2^ and a standard deviation (δ) of 284.69 kJ·m^−2^. Again, the results provided by prototype P02 had a confidence level of 90.73% of the divergences in the range of (−500 kJ·m^−2^; +500 kJ·m^−2^).

## 4. Conclusions

This paper presented a new low-cost electronic system for the real-time measurement of surface solar radiation. The proposed system was composed of a solar radiation sensor, a conditioning circuit, a low-power ESP8266 microcontroller, a unit to register data (datalogger) and an internet user interface application. It was autonomous and provided remote access to surface solar radiation levels based on IoT tools. The proposed solution was composed of an electronic hardware and software tool capable of operating with minimal human intervention. The system had a power consumption of 5 W·h, a mass of 400 g and an estimated cost of USD 200.

The proposed system was validated with experimental tests comparing the results with those provided by INMet for Campina Grande, Brazil. The experimental tests, carried out in areas with wireless network availability, indicated that the device provided irradiation levels with an accuracy greater than 96.0% and a confidence level greater than 90%. It showed an estimated error of ±500 kJ·m^−2^, equivalent to 138.89 W·m^−2^.

Commercial solutions that combine measurement irradiation with wireless data transfer technology are scarce and costly. The proposed architecture stands out from commercial solutions due to its low cost, flexibility, fast installation, modularity, automated data processing capacity and applicability in a wide range of research areas such as meteorological, environmental, and photovoltaic power generation, among others.

The proposed architecture also stands out from related studies for the model accuracy and the ability to connect to wireless networks, self-monitoring algorithms, and automated and cloud data processing capabilities.

## 5. Patents

This device is patent pending at the National Institute of Industrial Property under the number BR1020200199846.

## Figures and Tables

**Figure 1 sensors-21-03836-f001:**
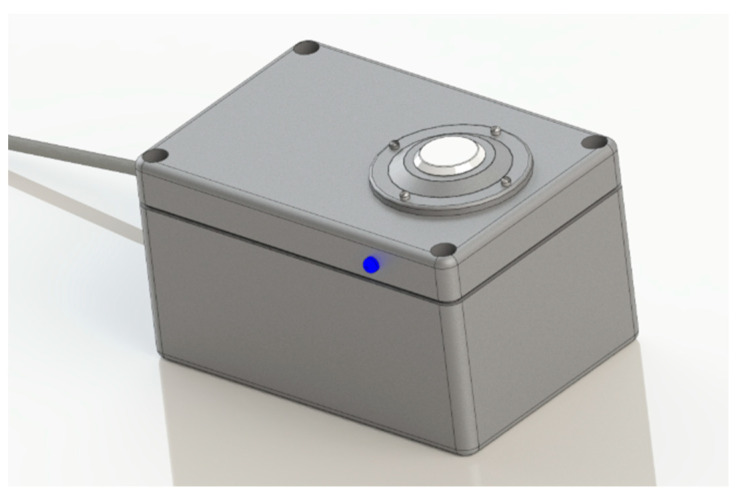
Concept developed in a virtual environment for the wireless solar irradiation monitoring system at the surface.

**Figure 2 sensors-21-03836-f002:**
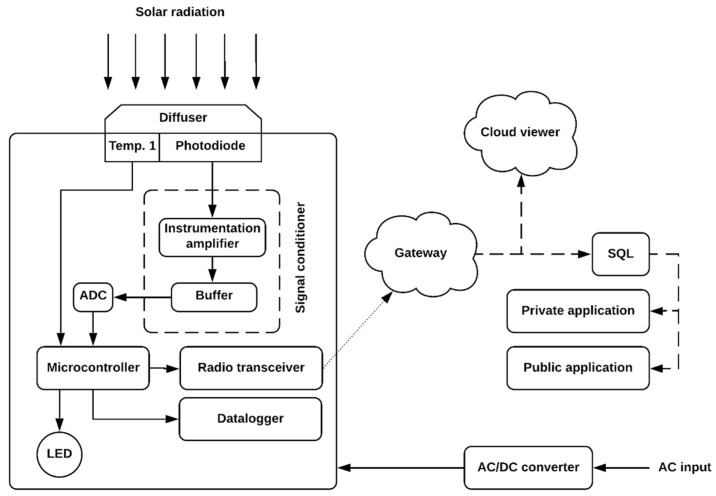
Block diagram of the wireless system for monitoring solar irradiation at the surface.

**Figure 3 sensors-21-03836-f003:**
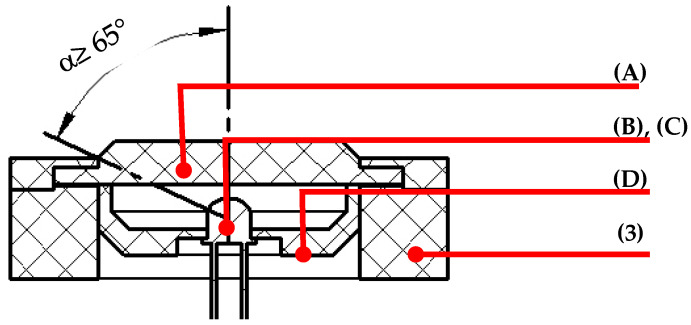
Sectional representation of the irradiation sensor architecture.

**Figure 4 sensors-21-03836-f004:**
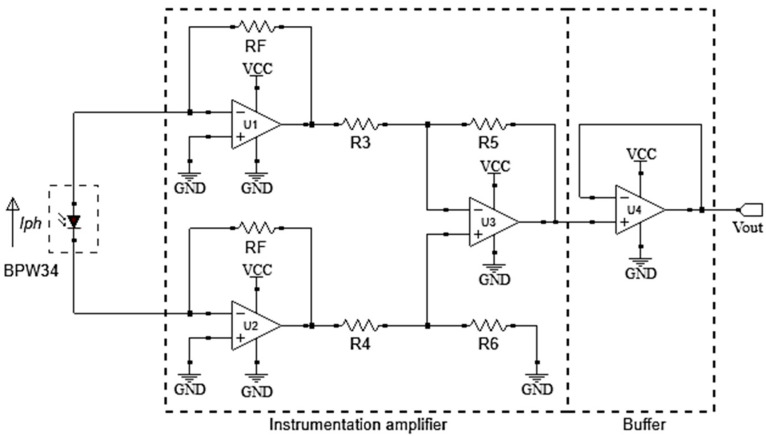
Architecture of the signal conditioning circuit.

**Figure 5 sensors-21-03836-f005:**
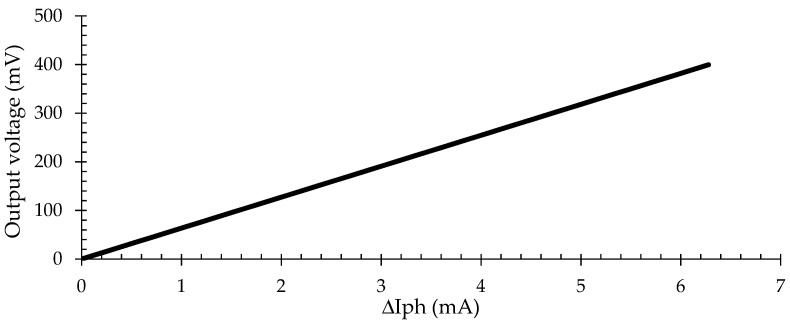
Output voltage (mV) behavior for the signal conditioning circuit.

**Figure 6 sensors-21-03836-f006:**
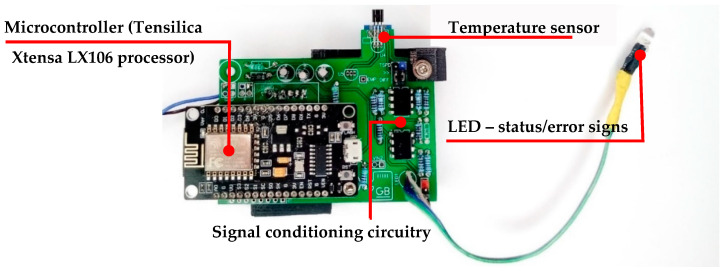
Prototype hardware for the real-time measurement of surface solar radiation.

**Figure 7 sensors-21-03836-f007:**
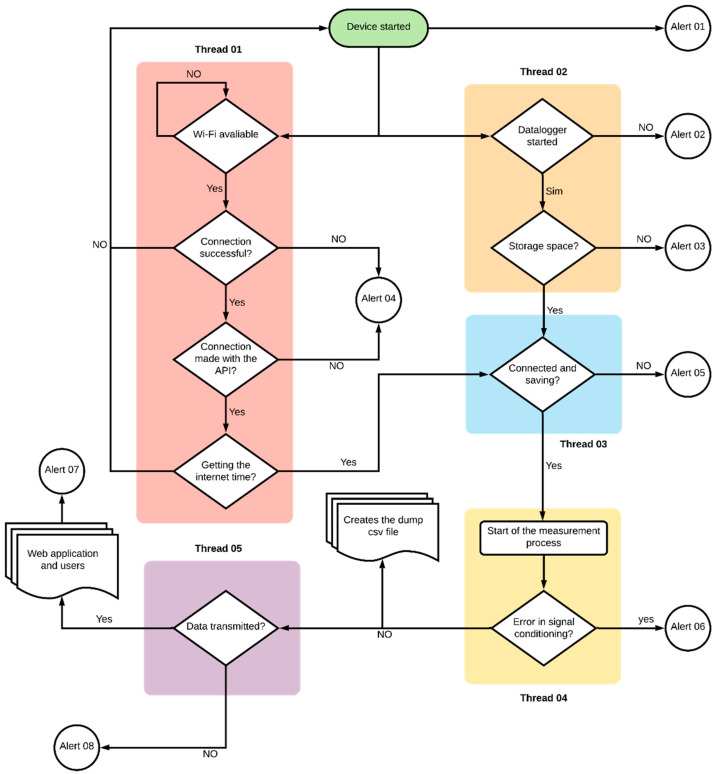
Flowchart of the identification of the firmware blocks and errors.

**Figure 8 sensors-21-03836-f008:**
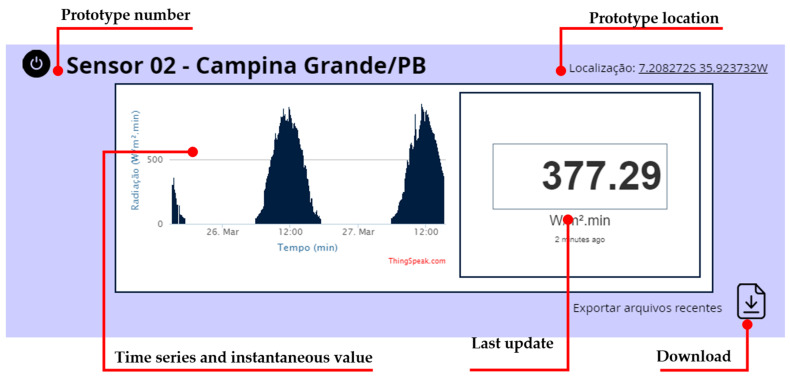
Data visualization web interface with its principal information.

**Figure 9 sensors-21-03836-f009:**
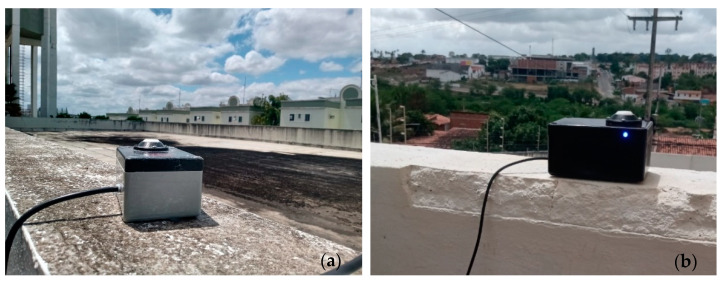
Prototypes installed in Campina Grande, Brazil: (**a**) P01, (**b**) P02.

**Figure 10 sensors-21-03836-f010:**
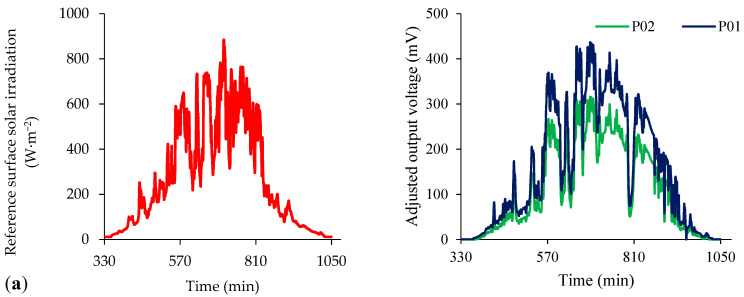
Reference surface solar irradiation (W·m^−2^) and voltage (mV) time series at a sensor output for low (**a**); moderate (**b**) and high (**c**) transmittance conditions.

**Figure 11 sensors-21-03836-f011:**
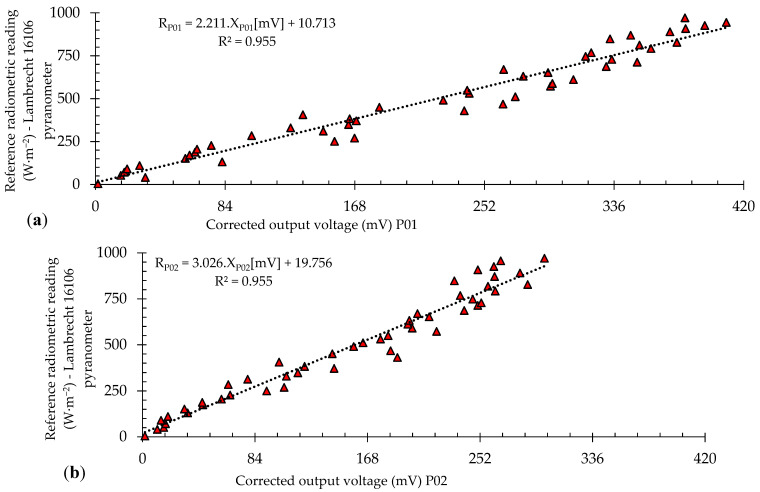
Solar irradiation by the reference pyranometer vs. the prototype response: (**a**) P01, (**b**) P02.

**Figure 12 sensors-21-03836-f012:**
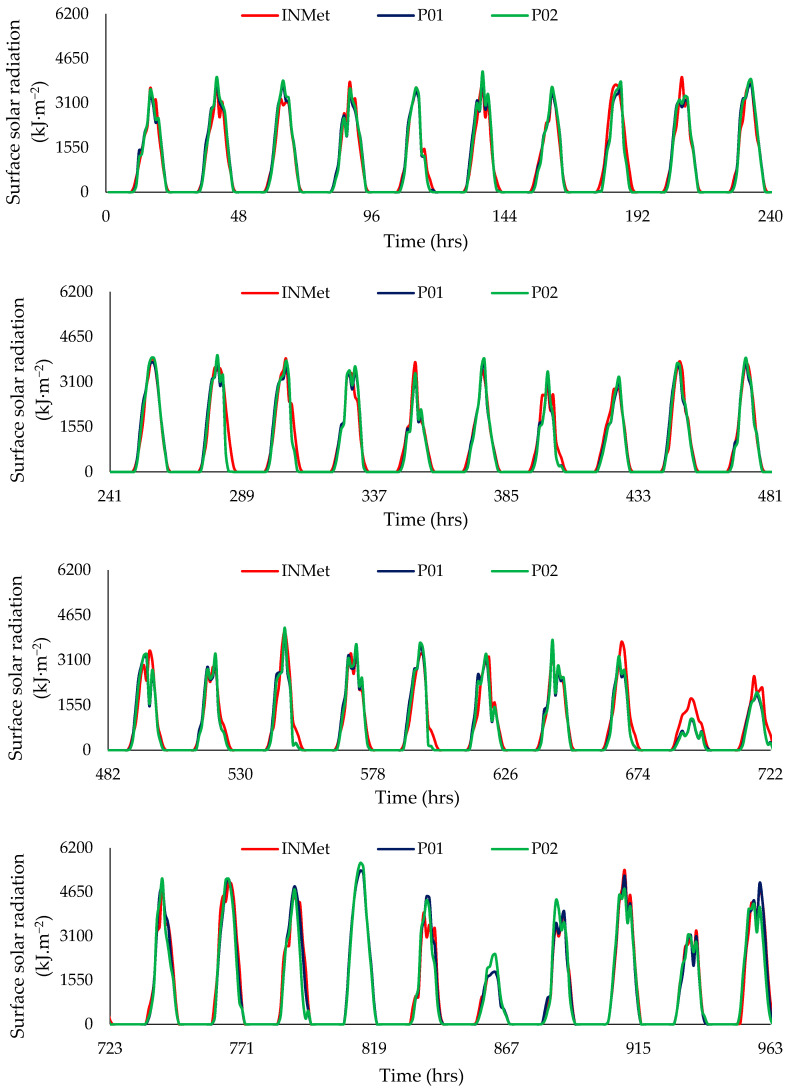
Comparison between the incoming surface solar radiation (INMet) and the responses of the prototypes (P01, P02) for Campina Grande, Brazil.

**Figure 13 sensors-21-03836-f013:**
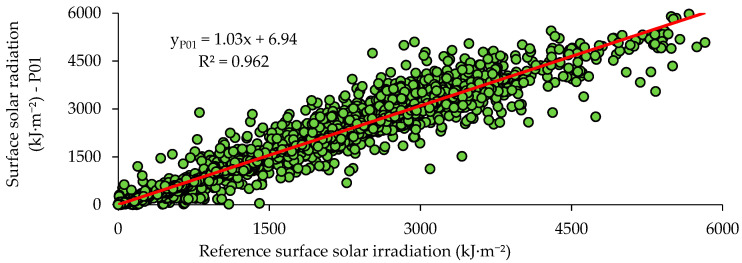
Scatter graph between the reference surface solar radiation (INMet) and the response of prototype P01 for Campina Grande, Brazil.

**Figure 14 sensors-21-03836-f014:**
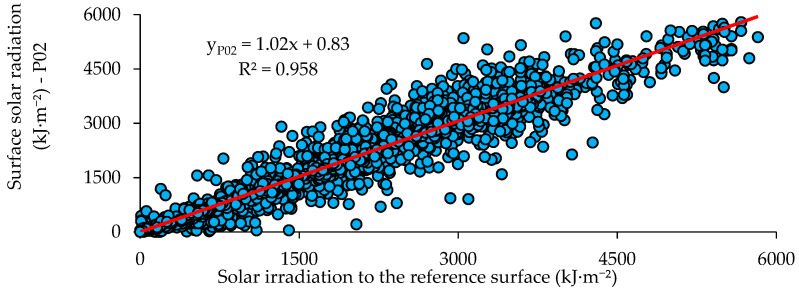
Scatter graph between the reference surface solar radiation (INMet) and the response of prototype P02 for Campina Grande, Brazil.

**Figure 15 sensors-21-03836-f015:**
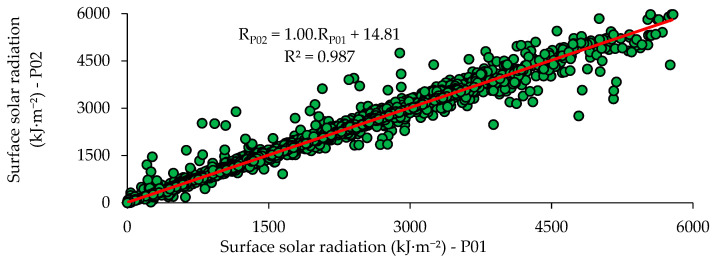
Scatter graph between the experimental results of prototype P01 and P02.

**Figure 16 sensors-21-03836-f016:**
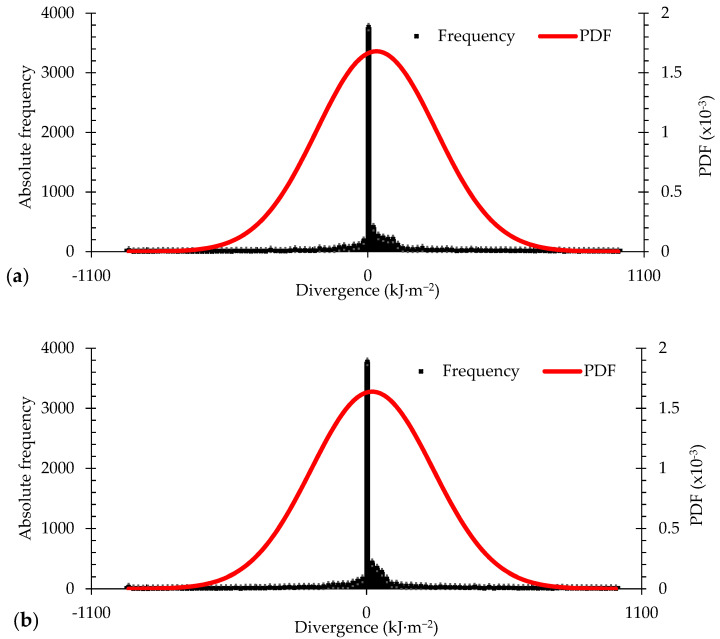
Relative frequency of the divergences for the prototypes: (**a**) P01, (**b**) P02.

**Table 1 sensors-21-03836-t001:** Basic features of bwp34 photodiode [24].

Technical Feature	Values
Radiation-sensitive area (mm^2^)	7.5
Sensitivity (A·W^−1^)	0.62
Noise equivalent power (W·Hz^−1/2^)	4 × 10^−14^
Spectral sensitivity (nm)	380 to 1400
Response time (ms)	0.1
Opening angle (°)	±65
Temperature coefficient (mV·K^−1^)	−2.6
Dimensions (mm)	5.4 × 4.3 × 3.2
Typical operating temperature (°C)	25

**Table 2 sensors-21-03836-t002:** Identification of alerts issued.

Alert	Meaning	Associated Color
1	Firmware startup	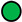	Green
2	Startup errors (datalogger)	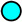	Turquoise blue
3	Datalogger—insufficient storage volume	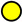	Yellow
4	Wireless connection failure	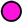	Pink
5	Datalogger or network connection failed	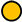	Orange
6	Sensor reading failure	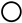	White
7	Full data transmission	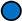	Blue
8	Data transmission failure	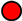	Red

**Table 3 sensors-21-03836-t003:** Pearson correlation coefficient performance classification (r) [47].

Pearson Correlation Coefficient (r)	Classification
0.91–0.99	Almost perfect
0.71–0.90	Too high
0.51–0.70	High
0.31–0.50	Moderate
0.11–0.30	Low
<0.10	Too low

**Table 4 sensors-21-03836-t004:** Performance rating for the confidence index (c) [47].

Confidence Index (c)	Classification
>0.85	Great
0.76–0.85	Very good
0.66–0.75	Good
0.61–0.65	Median
0.51–0.60	Poor
0.41–0.50	Bad
<0.40	Unsatisfactory

**Table 5 sensors-21-03836-t005:** Performance classification for the quadratic root of relative errors (rRMS) [51].

Quadratic Root of Relative Errors (rRMS)	Classification
>0.30	Poor
0.20–0.30	Acceptable
0.10–0.20	Good
<0.10	Excellent

**Table 6 sensors-21-03836-t006:** Summary of statistical indices for prototypes P01 and P02 after the calibration process.

Metrics	Prototype P01	Prototype P02
Coefficients of determination (R^2^)	0.955	0.955
Correlation coefficients (R)	0.977	0.977
Pearson coefficient (r)	0.977	0.977
Agreement index (d)	0.953	0.950
Confidence index (c)	0.932	0.928

## Data Availability

In our article, all the data are disclosed and explained in different parts of the article.

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
