# Peer review of "Development of a Real-Time Surface Solar Radiation Measurement System Based on the Internet of Things (IoT)"

_sensors, 2021, doi:10.3390/s21113836_

Round 1
Reviewer 1 Report
The paper proposes a novel (patent pending) solar radiation measurement system based on silicon photodiodes (Si) with a spectral range between 300 and 1400 nm, and incorporating Internet of Things (IoT) technology. Calibration and experimental research have been successfully conducted, proving good measurement performances of the proposed low-cost system.
The following issues are recommended to improve the paper:
- Abstract: avoid repeating the same word in the same sentence, e.g., agriculture in “The determination of the levels of solar radiation incident on the terrestrial surface (W.m-2) is essential for several areas such as architecture, agriculture, health, power generation, telecommunications, agriculture, and climate forecasting models”. Moreover, check carefully the paper for typing errors, e.g., “…can have a high ability capacity to…”, “that allows the photodiode a semi angle of sensitivity (a) greater” – probably (a) should be replaced by (α) – alpha, “Correted output voltage” instead of “Corrected…”, etc.
- Recommendation to extend the Introduction by describing additional relevant references on the paper topic, identify the knowledge limits and previous research drawbacks, the novelty promoted in your work and the structure of the paper. In its actual form, the Introduction is a very short presentation of the considered scientific domain.
- Figure 3: the figure depicts the sensor, not only the diffuser as suggested by its caption.
- Line 127: “based on irradiation of 100W.m-2 and photodiode characteristics listed in Table 1.” I suppose is about irradiance instead of irradiation (i.e, the radiant energyemitted into the surrounding environment (joule per square metre, J/m2) during that time period. This integrated solar irradiance is called solar irradiation, solar exposure, solar insolation, or insolation.)
- Strong recommendation to improve the English style, by a native speaker.
- As the proposed measurement system is a patent pending, please declare it explicitly in the paper and justify the novelty of the presented solution.
- Equation (5) contains wrongly the parameter α (alpha) instead of θ (theta), as used in Eq. (3).
- Figure 12 may be reduced to only several relevant days. In actual form, it is very extended (on 5 pages!) and useless for readers. Preferable to consider the measurement units declared previously and typically used in practice: Wm-2 instead of kJm-2h-1.
Author Response
Answers to the reviewer comments
We are very thankful for the reviewer's comments. They were essential to the improvement of our manuscript quality. Following it is the answer for all comments and the descriptions of the Page and Line where the correction can be found. All modifications are highlighted in the “red” color.
Reviewer 1:
Comments and Suggestions for Authors
The paper proposes a novel (patent pending) solar radiation measurement system based on silicon photodiodes (Si) with a spectral range between 300 and 1400 nm, and incorporating Internet of Things (IoT) technology. Calibration and experimental research have been successfully conducted, proving good measurement performances of the proposed low-cost system.
The following issues are recommended to improve the paper:
Abstract: avoid repeating the same word in the same sentence, e.g., agriculture in “The determination of the levels of solar radiation incident on the terrestrial surface (W.m-2) is essential for several areas such as architecture, agriculture, health, power generation, telecommunications, agriculture, and climate forecasting models”. Moreover, check carefully the paper for typing errors, e.g., “…can have a high ability capacity to…”, “that allows the photodiode a semi angle of sensitivity (a) greater” – probably (a) should be replaced by (α) – alpha, “Correted output voltage” instead of “Corrected…”, etc.
Answer: Done.
Recommendation to extend the Introduction by describing additional relevant references on the paper topic, identify the knowledge limits and previous research drawbacks, the novelty promoted in your work and the structure of the paper. In its actual form, the Introduction is a very short presentation of the considered scientific domain.
Answer: Done. Page 2, Lines 56-88.
Figure 3: the figure depicts the sensor, not only the diffuser as suggested by its caption.
Answer: Done.
Line 127: “based on irradiation of 100W.m-2 and photodiode characteristics listed in Table 1.” I suppose is about irradiance instead of irradiation (i.e, the radiant energy emitted into the surrounding environment (joule per square metre, J/m2) during that time period. This integrated solar irradiance is called solar irradiation, solar exposure, solar insolation, or insolation.)
Answer: Done.
Strong recommendation to improve the English style, by a native speaker.
Answer: Done.
As the proposed measurement system is a patent pending, please declare it explicitly in the paper and justify the novelty of the presented solution.
Answer: Done. Page 16, Lines 434-440.
Equation (5) contains wrongly the parameter α (alpha) instead of θ (theta), as used in Eq. (3).
Answer: Done.
Figure 12 may be reduced to only several relevant days. In actual form, it is very extended (on 5 pages!) and useless for readers. Preferable to consider the measurement units declared previously and typically used in practice: Wm-2 instead of kJm-2h-1.
Answer: Done. The system was tested for 200 days (4813 h) during the year 2020. However, it is shown only the first 40 days (963 h) since the pattern is repeated for the following days. We decided to keep the current unit because it is the best to represent the incoming solar radiation on the surface in an hour.

Reviewer 2 Report
The authors propose a prototype of solar radiation hardware based on silicon photodiodes (Si) supported for the IoT with an estimated cost of US$ 200.00.
It is necessary to review the keywords, inserting them in the title, abstract and other sections. It can increase your index visibility.
Is MQTT protocol a necessary keyword?
I suggest citing a recent article from Sensors.
Is it common to have a Wi-Fi connection in this environment? In the IoT scenario, other kinds of wireless technologies should be considered.
I felt a miss of related work section. It is crucial to present and compare your solution with others.
As the paper is more fit for the IIoT, I suggest comparing this solution with others from the marketplace.
In general, the paper shows a low-cost prototype patented by the authors. Considering the benefits and MAPE results and the clear presentation of the prototype, I believe that this paper is of interest to the Sensors' audience.
Author Response
Answers to the reviewer comments
We are very thankful for the reviewer's comments. They were essential to the improvement of our manuscript quality. Following it is the answer for all comments and the descriptions of the Page and Line where the correction can be found. All modifications are highlighted in the “red” color.
Reviewer 2:
Comments and Suggestions for Authors
The authors propose a prototype of solar radiation hardware based on silicon photodiodes (Si) supported for the IoT with an estimated cost of US$ 200.00.
It is necessary to review the keywords, inserting them in the title, abstract and other sections. It can increase your index visibility.
Answer: Done.
Is MQTT protocol a necessary keyword?
Answer: Done.
I suggest citing a recent article from Sensors.
Answer: There are only a few papers about this study topic in the Sensors journal. We used the following recently published article as an essential reference in our study:
Rocha, Á.B.d.; Fernandes, E.d.M.; Santos, C.A.C.d.; Diniz, J.M.T.; Junior, W.F.A. Development and Validation of an Autonomous System for Measurement of Sunshine Duration. Sensors 2020, 20, 4606. https://doi.org/10.3390/s20164606
Is it common to have a Wi-Fi connection in this environment? In the IoT scenario, other kinds of wireless technologies should be considered.
Answer: It is an important observation. Wi-Fi connection is commonly found in the experimental fields. However, the sensor has been developed to be applied in different sectors under several conditions. Thus, the MQTT (Message Queuing Telemetry Transport) protocol for data transmission was chosen because it is an OASIS standard messaging protocol for the Internet of Things (IoT). It is designed as an extremely lightweight publish/subscribe messaging transport that is ideal for connecting remote devices with a small code footprint and minimal network bandwidth. MQTT today is used in a wide variety of industries, such as automotive, manufacturing, telecommunications, oil, and gas, among others. It is known that short-range wireless technologies, e.g., Wireless Fidelity (Wi-Fi) (https://www.wi-fi.org/) for IoT applications in the energy sector, have been widely studied. However, due to the high power requirements of Wi-Fi, this technology is not the best solution in the energy sector. Low power wide area network (LPWAN) communication technologies such as narrowband IoT (NB-IoT); ZigBee; Bluetooth low energy (BLE) technologies; as well as the emerging LPWAN technologies such as LoRa, Sigfox, and LTE-M operating in the unlicensed band are better solutions to be used in the energy sector. Because, these emerging LPWAN technologies enable establishing a reliable, low-cost, low-power, long-range, last-mile technology for smart energy management solutions. Therefore, only the Wi-Fi connection was used for the purpose of this study, which was to develop a real-time surface solar radiation measurement system based on the Internet of Things (IoT).
I felt a miss of related work section. It is crucial to present and compare your solution with others.
Answer: Done. Page 12, Lines 350-351; Page 13, Lines 369-371; Page 14, Lines 375-376. However, there are only a few studies to compare in the scientific literature.
As the paper is more fit for the IIoT, I suggest comparing this solution with others from the marketplace.
Answer: Excellent point of view about our study. This study aimed to develop, implement, and validate a system that allows remote measurement of surface solar irradiation in real-time. The proposed sensor uses IoT technology to safely enable low-cost, intelligent, and reliable measures and provide radiometric data from multiple Web platforms. However, in general, the IIoT solutions found in the marketplace are related to the energy sector, i.e., energy supply, transmission and distribution, and demand. However, there are only a few studies to compare in the scientific literature. Additionally, in the manuscript, the comparisons are found on Page 12, Lines 350-351; Page 13, Lines 369-371; Page 14, Lines 375-376.

Round 2
Reviewer 1 Report
No any additional recommendations.